# Degradation of 5-Dialkylamino-Substituted Chlorsulfuron Derivatives in Alkaline Soil

**DOI:** 10.3390/molecules27051486

**Published:** 2022-02-23

**Authors:** Lei Wu, Yu-Cheng Gu, Yong-Hong Li, Fan-Fei Meng, Sha Zhou, Zheng-Ming Li

**Affiliations:** 1State Key Laboratory of Elemento-Organic Chemistry, College of Chemistry, Nankai University, Tianjin 300071, China; wl836929871@163.com (L.W.); bioassay@nankai.edu.cn (Y.-H.L.); ffm2017@126.com (F.-F.M.); 2Syngenta Jealott’s Hill International Research Centre, Bracknell, Berkshire RG42 6EY, UK; yucheng.gu@syngenta.com

**Keywords:** sulfonylurea herbicides, chlorsulfuron, soil degradation, alkaline soil, DT_50_

## Abstract

Sulfonylurea herbicides are widely used as acetolactate synthase (ALS) inhibitors due to their super-efficient activity. However, some sulfonylurea herbicides show toxicity under crop rotation due to their long degradation time, for example, chlorsulfuron. Our research goal is to obtain chlorsulfuron-derived herbicides with controllable degradation time, good crop safety and high herbicidal activities. Based on our previously reported results in acidic soil, we studied the degradation behaviors of 5-dialkylamino-substituted chlorsulfuron derivatives (**NL101**–**NL108**) in alkaline soil (pH 8.39). The experimental data indicate that addition of the 5-dialkylamino groups on the benzene ring of chlorsulfuron greatly accelerated degradation in alkaline soil. These chlorsulfuron derivatives degrade 10.8 to 51.8 times faster than chlorsulfuron and exhibit excellent crop safety on wheat and corn (through pre-emergence treatment). With a comprehensive consideration of structures, bioassay activities, soil degradation and crop safety, it could be concluded that 5-dialkylamino-substituted chlorsulfuron derivatives are potential green sulfonylurea herbicides for pre-emergence treatment on both wheat and corn. The study also provides valuable information for the discovery of new sulfonylurea herbicides for crop rotation.

## 1. Introduction

Sulfonylureas were found to have super herbicidal activities in 1987 and have been used worldwide due to their high activity and low toxicity to mammals [1]. They are acetolactate synthase (ALS) inhibitors and can block the first reaction in the pathway of branched-chain amino acid biosynthesis in weeds [2].

In order to meet the large demand of food for the growing population, crop rotation mode has been implemented in some countries, including China [3]. For example, the wheat–corn rotation mode is widely used in northern China. However, different sulfonylurea herbicides show different toxicities on different crop varieties. Especially, the widely used chlorsulfuron exhibits good safety on wheat through pre-emergence treatment, but the residue in soil causes a certain degree of damage to the subsequent corn. The large-scale use of sulfonylurea herbicides has caused serious economic losses [3,4,5]; thus, in 2014, the Ministry of Agriculture of China suspended the field application of chlorsulfuron [6].

Previous studies have demonstrated that the soil degradation of sulfonylurea herbicides is pH dependent [7,8,9,10]. Fredrickson et al. reported that the DT_50_ (half-life of degradation) of chlorsulfuron was 12.5 weeks in soil with pH 8.0 and 1.9 weeks at pH 5.6 [8]. Thirunarayanan et al. reported that the DT_50_ of chlorsulfuron was 88.5 days in soil with pH 6.2 at 20 °C and 144 days at pH 8.1 [9]. Walker et al. also reported that the DT_50_ of chlorsulfuron was 22 days in soil with pH 5.6 and 124 days at pH 7.4 [10]. The results show a strong negative correlation between DT_50_ and pH [10]. In addition, sulfonylurea herbicides have a long residual period in alkaline soils.

From the point of view of “ecology-friendly pesticides”, the development of herbicides with controllable degradation is quite important.

In our previous study, we found that 5-substituents on the benzene ring of chlorsulfuron had a critical influence on their degradation rates in soils. In 2016, Li firstly concluded that electron-donating substituents accelerated the degradation, while the electron-withdrawing ones prolonged the degradation time in acidic soil [11,12]. In order to further study the substitute-degradation relationship, Ma and Li introduced various N-methylamido and dialkylamino substituents on the 5-position of the benzene ring in chlorsulfuron and studied the structure-DT_50_ relationship in soil with pH 5.52 [13].

The soil in most of northern China is alkaline (pH 7.5–8.5), and sulfonylureas degrade slowly and will cause harm to other subsequent crops [10]. In 2018, Li studied the degradation of dimethylamino- and diethylamino-substituted chlorsulfuron derivatives **NL101** and **NL106** in alkaline soil (pH 8.46) and showed that the degradation rate of **NL101** and **NL106** was nearly 15–30 times faster than that of chlorsulfuron, which was too fast for practical use [14].

It was found that most 5-dialkylamino-substituted chlorsulfuron derivatives maintained high herbicidal activities against both dicotyledons and monocotyledons through pre- and post-emergence treatment (Figure 1) [13]. However, their alkaline soil degradation and crop safety on wheat and corn have not been studied systematically.

In order to obtain sulfonylurea herbicides with good crop safety and a controllable degradation time for wheat–corn rotation in alkaline soil, 5-dialkylamino-substituted chlorsulfuron derivatives that maintain high herbicidal activities were selected and systematically studied under alkaline conditions.

## 2. Materials and Methods

### 2.1. Instruments and Materials

All reagents for high-performance liquid chromatography (HPLC) were chromatographic grade, reaction reagents were analytical grade and the water phase was double-distilled water. TU-1810 ultraviolet–visible spectrophotometer (Persee General Analysis Co., Beijing, China) was used to detect wavelength. HPLC data were obtained on a SHIMADZU LC-20AT (SHIMADZU Co., Kyoto, Japan), equipped with a binary pump (Shimadzu, LC-20AT), a UV–vis detector (Shimadzu, SPD-20A), an auto sampler (Shimadzu, SIL-20A), a Shimadzu shim-pack VP-ODS column (5 μm, 250 × 4.6 mm, C18 reversed phase chromatography) connected to a Shimadzu shim-pack GVP-ODS (10 × 4.6 mm) guard column, a column oven (Shimadzu, CTO-20AC) and a computer (model Dell) for carrying out the experimental data analysis. Column chromatography purification was carried out using silica gel (200–300 mesh). A SHZ-88 thermostatic oscillator (Jintan Medical Instrument Factory, Changzhou, China), Thermo Scientific Legend Mach 1.6 R centrifuge (Thermo Fisher Scientific Inc, Waltham, MA USA) and SPX-150B-Z biochemical incubator (Boxun Industrial Co., Shanghai, China) were used in the degradation experiment.

### 2.2. Compounds NL101-NL108

The structure of 5-substituted chlorsulfuron compounds is shown in Figure 2.

The synthetic procedure of 5-dialkylamino-substituted compounds is shown in Figure 3, which was reported in our previous papers [11,12,13].

### 2.3. Soil Degradation Assay

The methods for the analysis of soil degradation were reported in previous studies [13,14,15,16]. The detailed experimental procedure, including soil selection, HPLC conditions, the establishment of standard curve, measurement of the recovery rate and the cultivation of samples, can be found in the Appendix A.

Soil degradation steps are briefly described here. Soil with pH 8.39 was selected from Cangzhou, Hebei Province, to develop degradation studies [15]. Additionally, the properties of tested alkaline soil are listed in Table 1. For the analysis of target compounds by HPLC, chromatographically pure methanol, acetonitrile and ultrapure water (pH 3.0) were used as the mobile phase. The standard curves were established at 20 °C with a concentration range between 200 and 0.025 ug·mL^−1^. In addition, the retention time was no more than 20 min. The concentrations of the test compounds in the conical flask were 0.5, 2 and 5 mg·kg^−1^ (in acetonitrile) for 20 g of soil. Each concentration of sample was repeated 5 times. The recovery rates remained between 70 and 110% with a coefficient of variation less than 5%. For cultivation of samples, 3.5 mL of water was added to adjust 60% of the holding capacity of the sample. The concentration of each sample was 5 mg·kg^−1^ in soil and the samples were cultivated in the dark with 80% humidity at 25 ± 1 °C by a biochemical incubator. Soil samples of three replicates were collected at six different times. Finally, DT_50_ values were calculated according to the formula: DT_50_ = ln2/k. The analytical data for alkaline soil degradation are listed in Table 2 and Table 3.

The analytical data for verification of recovery rates in various concentrations are listed in Table 2.

### 2.4. Crop Safety Assay

Chlorsulfuron is a crop-selective sulfonylurea herbicide for wheat fields, but its soil residue seriously affects the growth of seedlings to the subsequent crops. We selected corn as the subsequently planted crop after wheat and studied the safety of synthesized compounds on wheat (Xinong 529) and corn (Xindan 66). The culture method was consistent with previous reports [13,16].

Methods of plant cultivation: Artificially mixed soil (loam, vermiculite and fertilizer soil (v/v/v = 1:1:1)) was packed into paper cups (250 mL) with a diameter of 7.0 cm. The crop seeds (0.6 cm deep) were planted in the mixed soil. The cups were covered with plastic wrap to keep them moist until the plants sprouted, and plants were grown at 25 ± 1 °C in the green house. Plants were watered regularly to ensure the normal growth of the crops.

Wheat safety assay: The wheat was tested with the target compound through pre- and post-emergence treatment via pot trials at the concentration of 30 and 60 g·ha^−1^. The fresh weight of the cover crops was measured after 22 days for pre-emergence treatment. For post-emergence treatment, the safety assay began when the wheat grew to the 4-leaf stage. After spraying the compounds, the fresh weight of the cover crops was measured after 28 days for post-emergence treatment.

In the case of corn, the detailed crop safety assay was consistent with wheat. The fresh weight of the cover crops was measured after 16 days for pre-emergence treatment. For post-emergence treatment, the safety assay began when the wheat grew to the 3-leaf stage. After spraying the compounds, the fresh weight of the cover crops was measured after 23 days for post-emergence treatment.

The fresh weight of the cover crops was measured after several days, and the inhibition rates of fresh weight were used to represent the safety of crops. The data were analyzed through Duncan multiple comparison by SPSS 22.0.

## 3. Results

### 3.1. Soil Degradation Results

The degradation of compounds was firstly examined in alkaline soil (pH 8.39) with chlorsulfuron as the control. The degradation curves of the first-order kinetic equation of tested compounds were established according to the data from six samplings. The DT_50_ of the test compounds was calculated, as shown in Table 3.

As shown in Table 3, the degradation data from alkaline soil (pH 8.39) indicate that the DT_50_ of **NL101–NL108** was reduced to 3.03–14.6 days, while the DT_50_ of chlorsulfuron was 158 days (Figure 4).

The degradation time of **NL101–NL108** was 10.8–51.8 times faster than that of chlorsulfuron. The DT_50_ of **NL101** was 3.03 days, which was 51.8 times faster than that of chlorsulfuron. For the other compounds, DT_50_ of **NL102** was 5.66 days (27.9 times faster), DT_50_ of **NL103** was 7.74 days (20.4 times faster), DT_50_ of **NL104** was 14.6 days (10.8 times faster), DT_50_ of **NL105** was 6.38 days (24.8 times faster), DT_50_ of **NL106** was 6.39 days (24.7 times faster), DT_50_ of **NL107** was 8.20 days (19.3 times faster) and DT_50_ of **NL108** was 13.6 days (11.6 times faster).

### 3.2. Crop Safety Results

The crop safety of the target compounds is shown in Table 4 (for wheat) and Table 5 (for corn).

As the data show, the crop safety was screened through pot experiments at 30 and 60 g·ha^−1^ with chlorsulfuron as the control. The inhibition rates of **NL102**, **NL103**, **NL104** and **NL106**, presented in Table 4 and Table 5, indicate that these compounds were safe on wheat and corn through pre-emergence treatment.

## 4. Discussion

### 4.1. Soil Degradation Rates

In 1995, DuPont reported that flupyrsulfuron-methyl (DPX-KR-459) could accelerate degradation in alkaline soils [17]. Flupyrsulfuron-methyl degraded rapidly at 25 °C in pH 5–9 with a DT_50_ of 0.42–44 days [18]. Subsequently, iodosulfuron-methyl and foramsulfuron were also reported to exhibit faster degradation [19,20,21]. Tang et al. reported that the DT_50_ of iodosulfuron-methyl was 15.6 days in soil with pH 7.29 at 25 °C and 25.1 days at pH 9.42 [20]. Wu et al. reported that DT_50_ of foramsulfuron was 10.8 days in soil with pH 5.29 at 25 °C and 31.5 days at pH 7.86 [21]. Compared with chlorsulfuron, which has a longer degradation time, these sulfonylurea herbicides containing 5th substituents on the benzene ring exhibit faster degradation rates.

Chlorsulfuron degraded slowly in alkaline soil. Fredrickson et al. reported that the DT_50_ of chlorsulfuron in silty clay loam was 10 weeks at pH 7.5 and 12.5 weeks at pH 8.0 [8]. Thirunarayanan et al. reported that the DT_50_ of chlorsulfuron was 144 days in soil with pH 8.1 at 20 °C [9].

In this research, the degradation of **NL101**–**NL108** in alkaline soil was 10.8–51.8 times faster than that of chlorsulfuron. For 5-dialkylamino-substituted chlorsulfuron derivatives, any changes on the amino moiety will affect their soil degradation rates. For symmetric saturated alkane-substituted compounds, the DT_50_ of **NL101**, **NL106** and **NL107** was 3.03, 6.39 and 8.20 days, respectively. It seemed that the degradation half-life of the target compounds increases as the number of carbon atoms on the nitrogen atom increases. Additionally, the DT_50_ of the target compounds increased as the volume of 5-dialkylamino groups on the benzene ring of chlorsulfuron increased (e.g., DT_50_ of **NL101**, **NL102**, **NL106**, **NL107** and **NL108** was 3.03, 5.66, 6.39, 8.20 and 13.6 days, respectively). **NL102**, **NL103** and **NL106** contained the same number of carbon atoms bonded to the nitrogen atom, while the DT_50_ of **NL103** (7.74 days) was more than that of **NL102** (5.66 days) and **NL106** (6.39 days). It was speculated that saturated substituted structures might degrade faster than unsaturated substituted structures. Based on the alkaline soil degradation results, we could conclude that 5-dialkylamino-substituted chlorsulfuron derivatives could greatly accelerate the degradation in alkaline soil (with pH 8.39).

In 2020, Li reported that the DT_50_ of NL101-NL108 varied from 3.57 to 9.76 days in acidic soil (pH 5.52), which was 1.34–3.67 times faster than chlorsulfuron (13.1 days) [13].

A comparison of the degradation results of target compounds in acidic and alkaline soil is listed in Table 6.

Compared with the degradation rates in acidic soil, we found that 5-dialkylamino-substituted groups could greatly accelerate the degradation both in acidic and alkaline soil in comparison with chlorsulfuron. Moreover, the degradation speed in alkaline soil was faster than that in acidic soil.

### 4.2. Crop Safety

Chlorsulfuron is a popular sulfonylurea herbicide applied to wheat fields, but it causes a certain degree of damage to corn [22]. Iodosulfuron-methyl and foramsulfuron, kinds of sulfonylurea herbicides that contain 5th substituents on the benzene ring, are safe for the growth of crop seedlings in corn and grain fields [19,23,24]. It appears that 5th substituents on the benzene ring are potential herbicides to improve crop safety.

In 2018, Zhou reported that a dimethylamino-substituted chlorsulfuron derivative was less safe on wheat (Jima 22) but could improve safety on corn (Xindan 66) (Figure 5) [25].

In this research, 5-dialkylamino-substituted chlorsulfuron derivatives were tested for a crop safety assay on wheat and corn.

The inhibition rates of target compounds indicated that most compounds showed safety for wheat growth through pre-emergence treatment. At 30 g·ha^−1^, the inhibition rate of chlorsulfuron was 0%. For **NL102**, **NL103** and **NL104**, the inhibition rates were 2.1, 4.5 and 0%, respectively. On the contrary, they exhibited poor safety for post-emergence treatment.

In the case of corn, it was noted that the inhibition rates were greatly reduced through pre-emergence treatment. For the pre-emergence treatment of corn, at 30 and 60 g·ha^−^^1^, the inhibition rates of chlorsulfuron were 32.6 and 61.5%, respectively. The inhibition rate of **NL102** decreased from 32.6 and 61.5% to 0.4 and 5.6%, the inhibition rate of **N****L103** decreased to 0 and 8.7%, the inhibition rate of **NL104** decreased to 0 and 0% and the inhibition rate of **NL106** decreased to 0 and 4.7%, respectively. However, post-emergence treatment could seriously endanger the normal growth of the corn seedlings. It was speculated that 5-dialkylamino-substituted groups on the benzene ring could improve the crop safety of chlorsulfuron on corn through pre-emergence treatment.

Based on the above results, we believe that compounds such as **NL102**, **NL103** and **NL104** are potential green sulfonylurea herbicides for the pre-emergence treatment on wheat and corn.

## 5. Conclusions

Followed by our previously reported studies on herbicidal activities and acidic soil degradation, we systematically studied the degradation and crop safety of 5-dialkylamino-substituted chlorsulfuron derivatives in alkaline soil (pH 8.39) for the first time. We found that 5-dialkylamino-substituted groups on the benzene ring of chlorsulfuron could greatly accelerate the degradation both in acidic and alkaline soil in comparison with chlorsulfuron. Especially in alkaline soil, the degradation rate of the target compounds accelerated by 10.8–51.8 fold as compared with chlorsulfuron. Moreover, the DT_50_ of the target compounds increased as the number of carbon atoms bonded to the nitrogen atom increased. For crop safety, it was noted that 5-dialkylamino derivatives exhibited good crop safety through pre-emergence on wheat. Additionally, they could greatly improve the safety on corn compared with chlorsulfuron. We strongly believe that compounds such as **NL104** are potential green sulfonylurea herbicides for the pre-emergence treatment on both wheat and corn and will exhibit advantages in rotation mode. It is a preliminary exploration in this new field. Further research will include the synthesis of new sulfonylurea derivatives and studies of their bioassay activities, controllable degradation and crop safety. We hope to find new sulfonylurea herbicides that can be used on a wider range of crops and in crop rotation mode.

## Figures and Tables

**Figure 1 molecules-27-01486-f001:**
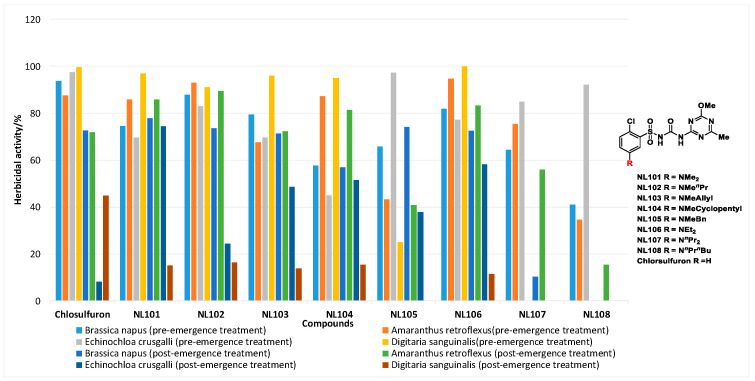
The herbicidal activity of 5-dialkylamino-substituted compounds against both dicotyledons and monocotyledons at 15 g·ha^−1^. (The full data can be found in Appendix A).

**Figure 2 molecules-27-01486-f002:**
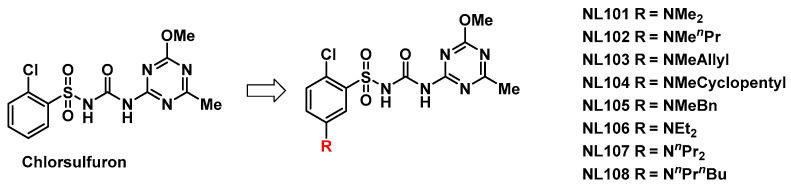
The Structure of 5-dialkylamino substituted compounds.

**Figure 3 molecules-27-01486-f003:**
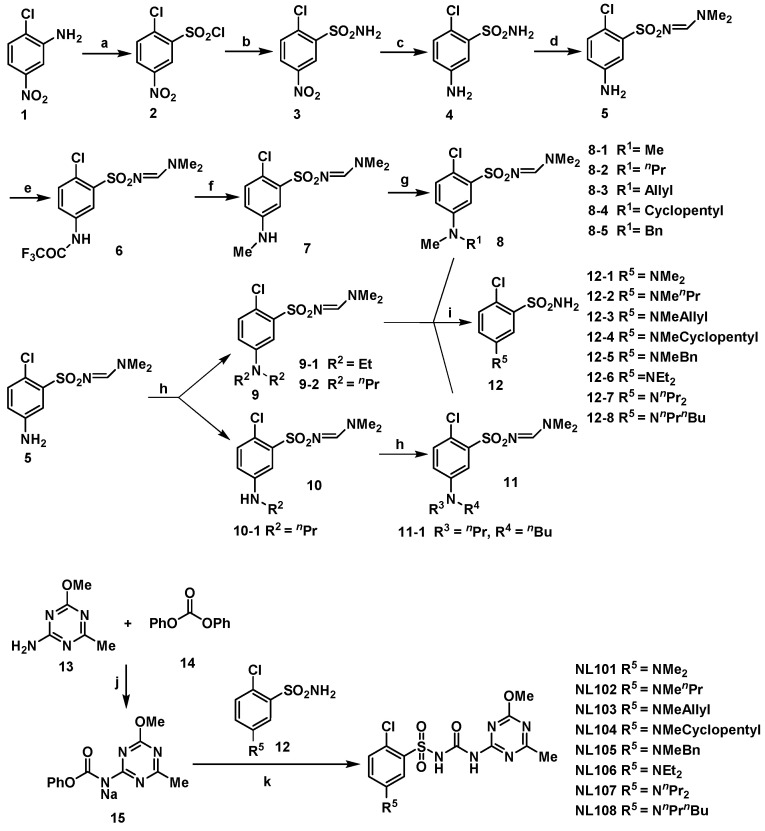
The synthesis of 5-dialkylamino-substituted compounds. Reagents and conditions: (a) H_2_O, HCl, NaNO_2_→H_2_O, HCl, CuCl_2_, NaHSO_3_, −5 °C; (b) 28% NH_3_·H_2_O, THF, 0 °C→RT (room temperature), overnight; (c) Fe, HCl, C_2_H_5_OH, H_2_O, reflux; (d) DMF-DMA (N,N-dimethylformamide dimethyl acetal), CH_2_Cl_2_; (e) TFAA (trifluoroacetic anhydride), CH_2_Cl_2_, 0 °C; (f) ICH_3_, K_2_CO_3_, DMF, 50 °C; (g) Haloalkane, K_2_CO_3_, CH_3_CN, reflux; (h) Haloalkane, K_2_CO_3_, CH_3_CN, reflux; (i) 80% H_2_NNH_2_·H_2_O, C_2_H_5_OH; (j) 60% NaH, THF, 0 °C→RT; (k) DBU (1,8-Diazabicyclo[5.4.0]undec-7-ene), CH_3_CN.

**Figure 4 molecules-27-01486-f004:**
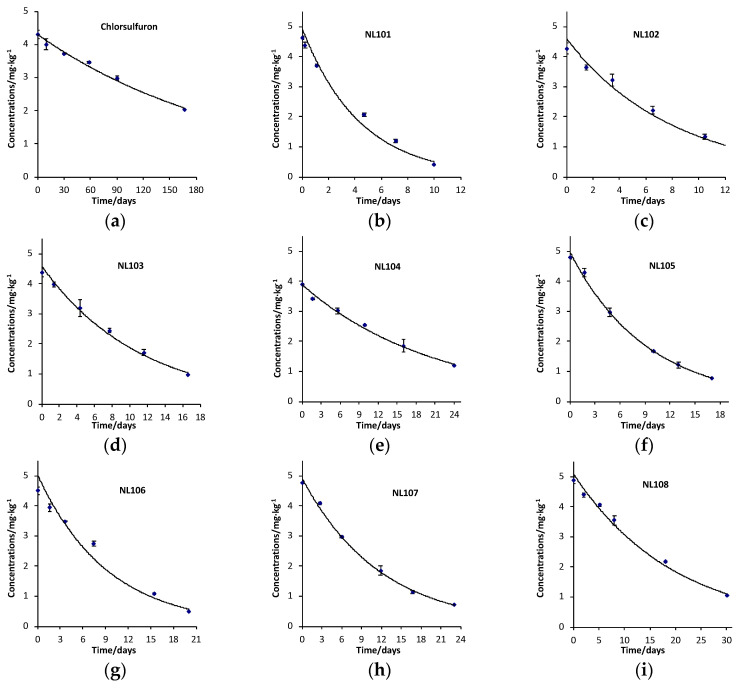
Degradation curve of target compounds (alkaline soil, pH 8.39). (**a**–**i**): Chlorsulfuron, NL101-NL108.

**Figure 5 molecules-27-01486-f005:**
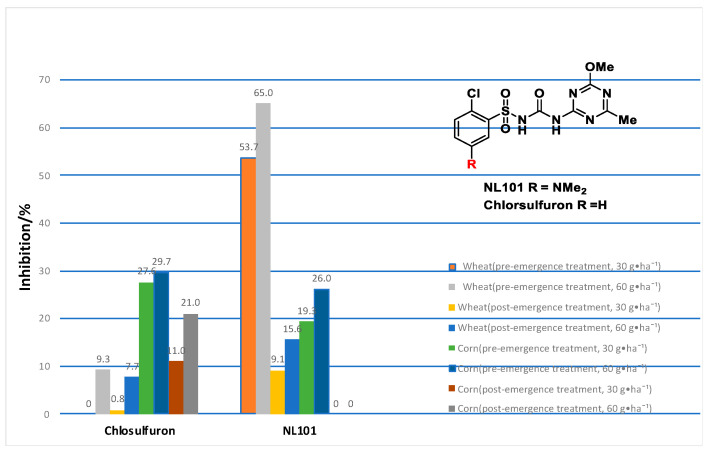
Crop safety of **NL101** on wheat (Jima 22) and corn (Xindan 66).

**Table 1 molecules-27-01486-t001:** Analytical data of soils.

Soils	Soil Texture	pH	Cation Exchange Capacity (cmol^+^·kg^−1^)	Organic Matter (g·kg^−1^)	Soil Separation (mm)/Mechanical Composition (%)								
Alkaline soils	loam	8.39	7.30	19.4	**1–2**	**0.5–1**	**0.025–0.5**	0.05–0.02	0.02–0.002	<0.002	0.25–0.05	2.0–0.05	0.05–0.002
					0.795	2.46	2.33	7.90	28.6	28.2	29.7	35.3	36.5

**Table 2 molecules-27-01486-t002:** Analytical data on the recovery rates of three concentrations (in soil with pH 8.39).

Compound	HPLC Analysis Condition (Wavelength, Flow Rate, Mobile Phase (v:v))	Extraction Solvent (v:v)	Additive Concentration (mg·kg^−1^)	Average Recovery Rate (%)	Coefficient of Variation RSD (%)
NL101	235 nm, 0.65 mL·min^−1^, CH_3_OH: H_3_PO_4_ (aq) (pH 3.0) = 60: 40	CH_3_COCH_3_: CH_2_Cl_2_: THF: H_3_PO_4_ (aq) (pH 1.5) = 30: 10: 10: 10	5	82.42	2.39
			2	72.52	1.94
			0.5	73.51	1.41
NL102	235 nm, 0.90 mL·min^−1^, CH_3_OH: H_3_PO_4_ (aq) (pH 3.0) = 78: 22	CH_3_COCH_3_: CH_2_Cl_2_: H_3_PO_4_ (aq) (pH 1.5) = 40: 5: 5	5	86.87	1.27
			2	84.24	2.34
			0.5	81.43	2.89
NL103	235 nm, 0.80 mL·min^−1^, CH_3_OH: H_3_PO_4_ (aq) (pH 3.0) = 75: 25	CH_3_COCH_3_: CH_2_Cl_2_: H_3_PO_4_ (aq) (pH 1.5) = 40: 5: 5	5	88.74	0.74
			2	87.24	0.83
			0.5	97.58	2.05
NL104	235 nm, 1.0 mL·min^−1^, CH_3_OH: H_3_PO_4_ (aq) (pH 3.0) = 77: 23	CH_3_COCH_3_: CH_2_Cl_2_: H_3_PO_4_ (aq) (pH 1.5) = 40: 5: 5	5	81.41	2.46
			2	89.43	1.93
			0.5	86.03	2.08
NL105	235 nm, 0.80 mL·min^−1^, CH_3_OH: H_3_PO_4_ (aq) (pH 3.0) = 78: 22	CH_3_COCH_3_: CH_2_Cl_2_: H_3_PO_4_ (aq) (pH 1.5) = 40: 5: 5	5	95.94	0.71
			2	99.87	1.14
			0.5	105.60	1.19
NL106	235 nm, 0.90 mL·min^−1^, CH_3_OH: H_3_PO_4_ (aq) (pH 3.0) = 77: 23	CH_3_COCH_3_: CH_2_Cl_2_: H_3_PO_4_ (aq) (pH 1.5) = 40: 5: 5	5	82.55	1.39
			2	85.47	2.19
			0.5	89.19	2.84
NL107	235 nm, 0.90 mL·min^−1^, CH_3_OH: H_3_PO_4_ (aq) (pH 3.0) = 80: 20	CH_3_COCH_3_: CH_2_Cl_2_: H_3_PO_4_ (aq) (pH 1.5) = 40: 5: 5	5	95.07	1.32
			2	90.97	1.31
			0.5	92.90	1.19
NL108	235 nm, 1.0 mL·min^−1^, CH_3_OH: H_3_PO_4_ (aq) (pH 3.0) = 82: 18	CH_3_COCH_3_: CH_2_Cl_2_: H_3_PO_4_ (aq) (pH 1.5) = 40: 5: 5	5	91.53	0.88
			2	91.39	0.89
			0.5	96.42	1.63
Chlorsulfuron	235 nm, 0.70 mL·min^−1^, CH_3_OH: H_3_PO_4_ (aq) (pH 3.0) = 62: 38	CH_3_COCH_3_: CH_2_Cl_2_: CH_3_OH: H_3_PO_4_ (aq) (pH 1.5): = 40: 5: 10: 10	5	73.54	1.09
			2	73.53	2.40
			0.5	81.09	1.16

**Table 3 molecules-27-01486-t003:** Kinetic parameters for alkaline soil (pH 8.39) degradation.

Compound	Kinetic Equations of Soil Degradation	Correlation Coefficient (R^2^)	DT_50_ (Days)
NL101	*C_t_* = 4.95e^−0.229t^	0.969	3.03
NL102	*C_t_* = 4.58e^−0.123t^	0.990	5.66
NL103	*C_t_* = 4.56e^−0.0895t^	0.991	7.74
NL104	*C_t_* = 3.88e^−0.0476t^	0.994	14.6
NL105	*C_t_* = 4.97e^−0.109t^	0.999	6.38
NL106	*C_t_* = 5.03e^−0.108t^	0.973	6.39
NL107	*C_t_* = 4.92e^−0.0845t^	0.997	8.20
NL108	*C_t_* = 5.10e^−0.0509t^	0.991	13.6
Chlorsulfuron	*C_t_* = 4.30e^−0.00440t^	0.990	158

**Table 4 molecules-27-01486-t004:** Crop safety of target compounds on wheat.

Compound	Concentration (g·ha^−1^)	Wheat (Xinong 529)								
		Pre. (22 Days after Treatment)					Post. (28 Days after Treatment)			
		Fresh Weight g/10 Strains	Analysis of Variance ^a^		Inhibition (%)	Fresh Weight g/10 Strains		Analysis of Variance ^a^		Inhibition (%)
			5%	1%				5%	1%	
	0	3.107	ab	AB	-	3.576		a	A	-
Chlorsulfuron	30	3.301	ab	A	0	3.323		a	A	7.1
	60	3.263	ab	A	0	2.152		b	B	39.8
NL102	30	3.041	ab	AB	2.1	0.57		def	CDE	78.8
	60	2.952	abc	AB	5.0	0.463		ef	DE	87.0
NL103	30	2.967	ab	AB	4.5	1.170		cde	BCDE	67.3
	60	2.786	abc	ABC	10.4	1.130		cde	BCDE	68.4
NL104	30	3.357	a	A	0	1.700		bc	BC	52.5
	60	3.165	ab	AB	0	1.601		bcd	BCD	55.2
NL106	30	2.095	def	CD	32.6	1.137		cde	BCDE	68.2
	60	2.025	def	CD	34.8	0.870		cdef	CDE	75.7

^a^ Among the averages, the same letter indicates that there was no significant difference, and different letters indicate that there was a significant difference.

**Table 5 molecules-27-01486-t005:** Crop safety of target compounds on corn.

Compound	Concentration (g·ha^−1^)	Corn (Xindan 66)								
		Pre. (16 Days after Treatment)					Post. (23 Days after Treatment)			
		Fresh Weight g/5 Strains	Analysis of Variance ^a^		Inhibition (%)	Fresh Weight g/5 Strains		Analysis of Variance ^a^		Inhibition (%)
			5%	1%				5%	1%	
	0	11.599	a	AB		9.214	a	A		-
Chlorsulfuron	30	7.813	b	BC	32.6	5.928	bc	BCD		35.7
	60	4.463	c	C	61.5	4.771	bcde	BCD		48.2
NL102	30	11.548	a	AB	0.4	5.146	bcde	BCD		44.1
	60	10.949	ab	AB	5.6	4.291	cde	BCD		53.4
NL103	30	12.590	a	A	0	5.813	bc	BCD		36.9
	60	10.593	ab	AB	8.7	3.571	de	CD		61.2
NL104	30	11.832	a	AB	0	6.517	b	B		29.3
	60	11.922	a	AB	0	5.620	bcd	BCD		39.0
NL106	30	11.770	a	AB	0	5.915	bc	BCD		35.8
	60	11.058	ab	AB	4.7	5.428	bcd	BCD		41.1

^a^ Among the averages, the same letter indicates that there was no significant difference, and different letters indicate that there was a significant difference.

**Table 6 molecules-27-01486-t006:** Comparison of acidic and alkaline soil degradation results of target compounds.

Compound	DT_50_ (days)	
	Acidic Soil (pH = 5.52)	Alkaline Soil (pH = 8.39)
NL101	3.57	3.03
NL102	5.06	5.66
NL103	5.78	7.74
NL104	8.45	14.6
NL105	9.00	6.38
NL106	7.30	6.39
NL107	9.76	8.20
NL108	7.45	13.6
Chlorsulfuron	13.1	158

## Data Availability

Data are contained within the article and Appendix A.

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
