# Peer review of "Degradation of 5-Dialkylamino-Substituted Chlorsulfuron Derivatives in Alkaline Soil"

_molecules, 2022, doi:10.3390/molecules27051486_

Round 1

Reviewer 1 Report

The article entitled "Research on the Controllable Alkaline Soil Degradation of 5 Dialkylamino Substituted Chlorosulfuron Derivatives" is an interesting piece of research focusing on the degradation of chlorosulfuron derivatives in alkaline soils. It reports the degradation and crop safety assays in alkaline soils of the previously referred herbicides. Although the research methods seem to be sound, the use of scientific literature to discuss the results (lines 145-187) is very limited to previous papers of the same authors who prepared this paper. Therefore, my opinion is that this paper needs to be refused, but has got high potential of being accepted once the discussion is improved with the required effort.

Line 10-14 - This is irrelevant for the scope of the article

Line 11 and subsequent lines - Why does chlorosulforon need capital letters?

Line 14-15 - This is irrelevant for the scope of the article

Graphs require removing top title and clear reference to the meaning of the variables in the axis in image footers. Footers and titles contain the same information.

Line 126-132 The presentation of results does not seek any insight into the correlation between chemical structure and degradation kinetics. This is another reason for me to reject this paper.

References - Of 17 references, only 12 are strictly scientific, peer-reviewed papers. 3 of them are self-citations.

Reviewer 2 Report

The manuscript is interesting considering the importance of data on pesticide degradation time in agricultural crop soils. So, it was employed analytical methods to investigate degradation time 5-dialkylamino-substituted chlrorosulfuron derivatives in soil of crop rotation mode. The authors have studied different aspects in acidic and alkaline soils. It is understandable, and it is well organized. I find no problems with the scientific approach or technical content presented by the authors in this manuscript.

Some suggestions:

p2, line 74: ‘5 µm’ not ‘5 mm’.

p2, line 76: prefer ‘guard column’ to ‘pre-column’.

p2, line 82 (Title): ‘Compounds NL101 – NL108’.

p4, Table 1: adjust the Table configuration. Numbers are out of their column positions.

p4, Table 2: coefficient of variation or relative standard deviation is represented by the symbol ‘%RSD’ not ‘R2’.

p6 and 7, Tables 4 and 5: ‘analysis of variance’.

Supplementary material (Soil degradation assay), 5° paragraph: ‘thermostatic’.

Reviewer 3 Report

The article is about Chlorsulfuron derivatives and there are several instances where this word have been replaced by the word Chlorosulfuron in the manuscript including the title which is confusing.

In the introduction, the objectives of the study are not fully explained on how to determine crop safety and insuring high herbicide activity of the Chlorsulfuron derivatives.

Author Response

Please see the attachment. Additionally, we have made revisions in the manuscript (in revised version).

Reviewer 4 Report

Mechanical composition (particle size distribution) should be given in % not g/kg
Why did you test only the loam, not other soil types?
Have you consider tests in other pH levels? Line 156 – where did you get that data?
Table 3 - please keep 3 main digits. 
Fig. 3 - I would recommend to keep tested NL in order (1-8). The line will not be that nice but it will be presented more clearly. 
Table 4 and 5 - what about NL101, 105, 107, 108?
Fig. 3 and 4 should be combined (and it is in Fig. 5). Data from figures are repeated in Table 6. Please show data only once. 
This work is connected with your other papers, however it looks just like a short communication not a scientific paper. I don’t like to look for the data reading about some results. In my opinion you should rewrite methods and show some more of already published data (even if it’s self-citation). 

Reviewer 5 Report

The presented paper titled: Research on the Controllable Alkaline Soil Degradation of 5-Dialkylamino Substituted Chlorosulfuron Derivatives prepared by authors Wu et al. shows interesting experimental data of Chlorsulfuron-derivated herbicides degradation behaviour in soil and crop safety.

Paper is prepared in good quality. Please modify following:

  • title - has to be changed-word research should be removed.
  • part introduction needs to be extended.
  • part discussion needs to be extended. The other studies with comparable results need to be discussed.
  • Table 2 is very complicated and some subscripts are incorrect. PLease modify and try to simplify
  • Part Materials and Methods- please add the tpe of wheat and corn used in experiments. Methods of plant cultivation are fully required.
  • Figures 3, 4 and 5 are based on single measurments ? Please try to correct and add SD.
  • authors did not explain how inhibition rate for plants was analysed
  • conclusions are too general.
  • text needs to be reviesed by english native speaker.

Author Response

Please see the attachment. In addition, we have undergone English language editing by MDPI.

Round 2

Reviewer 1 Report

Line 197-198-252- - " which seemed that when the number of carbon atoms on the nitrogen atom was the same"

Does this fragment refer to carbon atoms bonded to nitrogen atoms?

Reviewer 4 Report

Thank you for referring to the comments and for making appropriate corrections in the manustript. 

Reviewer 5 Report

Dear authors,

all my comments have been incorporated into revised version of manuscript. Therefore, I can just recommend paper for publication.
